# Authentication and Geographical Characterisation of Italian Grape Musts through Glucose and Fructose Carbon Isotopic Ratios Determined by LC-IRMS

**DOI:** 10.3390/molecules28031411

**Published:** 2023-02-02

**Authors:** Matteo Perini, Silvia Pianezze, Katia Guardini, Letizia Allari, Roberto Larcher

**Affiliations:** 1Centro Trasferimento Tecnologico, Fondazione Edmund Mach (FEM), Via E. Mach 1, 38098 San Michele all’Adige, Italy; 2Unione Italiana Vini Servizi (UIV), Viale del Lavoro 8, 37135 Verona, Italy

**Keywords:** stable isotope analysis, grape musts, LC-IRMS, authenticity, geographical traceability

## Abstract

The authenticity of grape musts is normally checked through a time-consuming stable isotopic analysis of carbon (*δ*^13^C) after fermentation and distillation by following the official OIV MA AS-312-06 method. In this study, the alternative use of a technique based on δ^13^C isotopic analysis of the major sugars of the grape must by liquid chromatography coupled with isotope ratio mass spectrometry (LC-IRMS) is provided. It allows not only the detection of the fraudulent addition to grape must of exogenous glucose and fructose deriving from C4 plants but also the characterisation of it based on its geographical origin. In order to discriminate between musts from different areas of Italy, a preliminary dataset was considered; the δ^13^C isotopic ratios of glucose and fructose of around 100 authentic samples were analysed. The two analysed parameters, ranging from −29.8‰ to −21.9‰, are well correlated (R^2^ = 0.7802) and the northern regions showed significantly more negative δ^13^C values for both sugars than the rest of the dataset.

## 1. Introduction

According to the International Organisation of Vine and Wine (OIV) definition, grape must is the beverage obtained from fresh grapes, whether naturally or through physical processes. The grapes, defined as the vine fruits designed for vinification, can also be overripe or slightly withered, as long as they can be pressed and generate a spontaneous alcoholic fermentation [1]. The raw product can be used to produce juices, whose actual alcoholic degree must be less than 1% vol, or it can be fermented to obtain wine.

Except for specific winegrowing regions and vintage areas, e.g., Brazil, Canada, Chile, China, France, Germany, Japan, New Zealand, Switzerland, United Kingdom, and the United States [2], the grape must cannot contain sugars other than those of grapes. In Italy, the addition of exogenous sugar (e.g., beet or cane) is forbidden and represents a fraud by unscrupulous producers to increase profit [3].

Since 1990, the OIV has adopted official methods to ensure the authenticity of must (now OIV MA-AS-311-05, MA-AS-312-06), based on the stable isotope ratio analysis of hydrogen (D/H, expressed in ppm) and carbon (^13^C/^12^C, expressed as *δ*^13^C) of the ethanol obtained after must fermentation [4]. In particular, *δ*^13^C analysis makes it possible to identify the addition of sugars from C4-plants such as cane or maize. These plants have a typical *δ*^13^C range of variability between −10‰ and −16‰, different from that of C3-plants such as grapes, which goes from −29‰ to −25‰ [5]. In addition, the use of the site-specific natural isotopic fractionation by the nuclear magnetic resonance (SNIF-NMR) technique allows one to determine the addition of beet sugar to the must. Indeed, grapes show D/H values for the methyl site of ethanol (ranging from 99 ppm to 105 ppm) different from the beet ones (ranging from 91 to 93 ppm) [6].

The same analytical techniques are normally used to build specific European reference databases according to European Community guidelines [4]. The databases are mainly used for the geographical traceability of must and wine [7]. Indeed, the two investigated parameters (*δ*^13^C and D/H) do not depend on the plant source only (grape, beet or cane), but also on the grape’s geographical origin. The different climatic conditions of the production areas, which depend on geographical parameters such as latitude and altitude, result in an isotopic variability in the plants and, therefore, in the ethanol obtained from the fermentation of the grape must [8]. The 2000–2010 Italian wine database reported by Dordevic et al. [9] is a clear example of this variability.

Unfortunately, the described methods present some issues that are difficult to overcome [10]. Grape must samples can only be analysed after they have been fermented to obtain ethanol. The process must be carried out under careful control of the fermentation to avoid the presence of unwanted by-products arising from a premature fermentation interruption [11]. Moreover, as indicated in the official OIV MA-AS-311-05 method, if the musts have been preserved by the addition of sulphur dioxide (SO_2_), they must undergo an additional step to eliminate the SO_2_, which would affect the fermentation. Once the product has been fermented, the ethanol must be separated using specific distillation columns (such as the Cadiot ones) making it possible to obtain ethanol free of isotopic fractionation [12] with a minimum alcohol degree of 95% vol. As these steps require high investment in people and time and necessitate the use of specific equipment, some developments are needed to find less time-consuming and expensive methods for the authentication of must, involving internal tracers for its traceability.

The direct analysis of the *δ*^13^C on the freeze-dried must is not advisable due to the presence of compounds other than sugars, such as organic acids, whose presence can affect the results [13]. On the other hand, the stable isotope ratio analysis of single chemical compounds, also named compound-specific isotope analysis (CSIA), provides a more in-depth understanding of a certain product than the traditional bulk analysis [14]. This technique can be performed through a gas chromatography combustion isotope ratio mass spectrometry (GC−C−IRMS) technique [15]. Perini et al. have recently proposed the analysis of the *δ*^13^C value of proline, analysed by GC−C−IRMS after extraction and derivatization, as a new potential geographical marker [16].

Since 2004, the year of its introduction on the market, the coupling of liquid chromatography with isotope ratio mass spectrometry (LC-IRMS) has shown its potential in the isotope analysis of matrices containing water-soluble compounds [17]. In LC–IRMS, analytes are separated on an LC system and consecutively oxidized in an online reactor to CO_2_, which is required for the determination of compound-specific carbon isotopic ratios. Reaction conditions in the interface depend on the flow conditions determined by the LC method and the flow rates and concentrations of oxidation agent and phosphoric acid added in the interface. This technique has been already used in the study of matrices such as wine [10], ethanol [10,12,18], glycerol [12], and honey [19,20] to detect fraudulent alterations of their natural composition such as the addition of exogenous sugars to the products. The LC-IRMS allows a single separation of the individual components of a sample and makes it possible to determine their δ^13^C values online, avoiding both the disadvantages of off-line methods and the disadvantages of methods requiring a derivatization step (such as GC-C-IRMS), causing the addition of extra carbons [18].

In this study, for the first time, the LC-IRMS technique was applied to the compound-specific study of the major sugars in Italian grape musts to first attempt to determine their authenticity and geographical origin. The *δ*^13^C analysis was performed on glucose and fructose of about 100 samples from 16 different Italian regions to characterise them based on their provenance. In addition, the *δ*^13^C_GLUCOSE_ and *δ*^13^C_FRUCTOSE_ variability in authentic and fake must (added with increasing percentages of exogenous sugars) has been explored and tested to verify their validity as fraud detectors.

## 2. Results and Discussion

### 2.1. Method Validation

The accuracy of the method was tested by comparing the *δ*^13^C_GLUCOSE_ and *δ*^13^C_FRUCTOSE_ of the standard solution resulting from the LC-IRMS with the values obtained through EA-IRMS for the same sugars. Differences between the results obtained thanks to the two techniques were ≤0.1‰ for both glucose and fructose. A within-day relative standard deviation (RSD) of repeatability of 2.5% for glucose and of 3.2% for fructose was calculated by analysing a selected grape must eight times. (Table 1). Moreover, to estimate a between-day relative standard deviation of repeatability, the same sample was also measured on three different days obtaining an RSD of 2.5% for glucose and of 1.3% for fructose (Table 1).

### 2.2. Geographical Characterization of Authentic Samples

The *δ*^13^C results of the LC-IRMS analysis for glucose and fructose, divided by the origin of the sample (whether north, centre or south Italy) are reported in Figure 1.

The *δ*^13^C_GLUCOSE_ ranges between −29.8 and −24.8‰ (average −27.6‰), −28.2 and −22.3‰ (average −25.0‰), and −28.9 and −22.8‰ (average −25.8‰) for northern, central and southern Italian regions, respectively. The δ^13^C_FRUCTOSE_ ranges between −29.6 and −24.0‰ (average −26.8‰), −28.5 and −21.9‰ (average −24.8‰), and −27.7 and −21.9‰ (average −25.4‰) for northern, central and southern Italian regions, respectively. This may be likely due to the influence of environmental factors on the *δ*^13^C of plants. Indeed, as high humidity results in high C isotope fractionation during plant biosynthesis, the δ^13^C of humid regions has been reported to be more negative than arid ones [21]. Plants normally exhibit higher δ^13^C values when high temperatures, low air humidity, and a high ground-water deficit lead to narrower stomatal apertures in the leaves of plants [22].

According to the one-way ANOVA, resulting in different letters for the statistically different groups, both *δ*^13^C_GLUCOSE_ and δ^13^C_FRUCTOSE_ made it possible to discriminate the regions of northern Italy (letter a, Figure 1) from the rest of the dataset (*p* < 0.01). On the contrary, the samples of central and southern Italy (letter b, Figure 1) were not statistically different from each other (*p* > 0.01). The samples from Basilicata and Campania (Appendix A, Appendix A) show relatively low *δ*^13^C values, in disagreement with the other samples from southern Italy, probably due to the late ripening of the grapes selected during the sampling in October 2021 (Figure 2). In fact, the variation of the *δ*^13^C is correlated with the climatic conditions before harvesting (the so-called “summer water stress”). Gaudillère et al. showed that the *δ*^13^C of must sugars decreased if conditions during berry maturation varied from dry to wet, as it normally happens by passing from an early harvest in August–September to a late one in October–November [23].

The *δ*^13^C_GLUCOSE_ and *δ*^13^C_FRUCTOSE_ proved to be positively correlated parameters (r^2^ = 0.7802, y = 0.8482x − 3.5178) as shown in Figure 2. The correlation is likely due to the strict connection between glucose and fructose metabolism in grapes during ripening [24]. In fact, during berry ripening, sucrose is translocated from leaves to fruit and rapidly converted to glucose and fructose [25].

The cluster of northern Italy samples is displayed in the left hand of the biplot (coloured circles, Figure 2), corresponding to relatively low *δ*^13^C_GLUCOSE_ and *δ*^13^C_FRUCTOSE_. On the other hand, central and southern Italy clusters (coloured triangles and rhombuses, Figure 2) are quite completely overlapped. This finding agrees with the results of the one-way ANOVA, highlighting the statistical difference between the groups.

The correlation between the *δ*^13^C of grape must sugars and ethanol deriving from its fermentation is reported in the literature [26,27]. Therefore, the data obtained might also be used to build the official isotope data banks, which must be built every year by member countries, as required by European legislation [4]. As reported by Perini et al., there is a constant difference of + 1.1‰ between the *δ*^13^C value of sugars and ethanol in wine, due to a non-equivalent distribution of ^13^C on the sugars’ carbonate skeleton. Indeed, carbons in position 3 and 4 in the glucose structure, which are ^13^C-enriched, are lost as CO_2_ during fermentation [28].

Potentially, the data obtained on the grape musts in LC-IRMS, once corrected for this constant deviation, would make it possible to obtain the *δ*^13^C values of ethanol without needing the time-consuming fermentation and distillation prescribed in the official OIV MA AS-312-06 method. Nevertheless, this explorative study, aiming to show the applicability of the described method, was carried out using a preliminary dataset, which will have to be improved to be converted into a databank.

### 2.3. Authentication of Grape Musts

To verify the power of the LC-IRMS approach for the authentication of grape must, authentic and fake samples added with increasing percentages of exogenous glucose and fructose were analysed.

The *δ*^13^C_GLUCOSE_ and *δ*^13^C_FRUCTOSE_ of all authentic musts ranged between −29.8 and −21.9‰, being the two parameters correlated to each other. The *δ*^13^C obtained are perfectly in line with the botanical origin from grape, a C3 plant that normally ranges from −30 to −23‰ [5].

As a method to confirm the authenticity of the must samples, the ratio between the *δ*^13^C of glucose and fructose (R_13C_G/F_), reported by Guyon et al. for sweet wine was considered [10]. The same ratio was also calculated on adulterated samples with exogenous glucose or fructose. While the authentic samples show an R_13C_G/F_ = 1.00 ± 0.04, consistent with the limit proposed by Guyon et al. (R_13C_G/F_ = 1.01 ± 0.03), the adulterated ones deviate from this interval [10]. Indeed, the addition of adulterant glucose to the authentic sample will change the *δ*^13^C_GLUCOSE_ only, resulting in an R_13C_G/F_ different from 1. The same will happen when adding fructose as an adulterant.

The results reported in Figure 3 show that the addition of glucose and fructose from C4 plants such as cane or maize increases the *δ*^13^C of both sugars in the grape must. As previously mentioned, this is due to C4 plants’ characteristic *δ*^13^C values, which normally range between −14 and −12‰ [5,29]. The difference between C3 and C4 plants’ values is mainly due to the photosynthetic pathways the plants follow during CO_2_ fixation, resulting in different *δ*^13^C [30]. During this process, each plant type (C3, C4) discriminates differently between the heavier and the lighter isotopes of carbon, resulting in different *δ*^13^C plant tissues. The reported results confirm the potential of this technique to discriminate between authentic grape must and must adulterated with exogenous sugars and to provide *δ*^13^C that might be used as an alternative for the building of a national database.

## 3. Materials and Methods

### 3.1. Reagents

The solutions of the oxidant reagent (sodium persulphate, Honeywell Fluka, ≥99.0%) and the acid one (orthophosphoric acid, Honeywell Fluka, ≥99.0%) were both prepared in milli-Q water (Arium Ultra-pure Lab Water System, Sartorius Stedim, Göttingen, Germany). The D- (+)-Glucose and D- (−)-Fructose used to prepare the standard solution were both Sigma Aldrich products, St. Louis, MO, USA. The reagents and the milli-Q water used as a solvent in the UHPLC system were previously degassed in a sonic bath for 30 min each.

### 3.2. Samples

The picking of the grapes was carried out directly in the field during the harvest (see date of harvest in Appendix A) by technicians of the various regions. The five withdrawal points of each field parcel were sited at the four corners and in a central point of it. At each sampling point, one or two bunches of grapes were collected, together representing an average sample. This was processed into grape must and 1 g/L of sodium azide was added to avoid the fermentation to start. A total of 95 authentic Italian must samples were considered. The 16 sampling regions were located in the north (Emilia-Romagna (7), Friuli-Venezia Giulia (8), Lombardy (8), Piedmont (7), Trentino-Alto Adige (5), Veneto (7)), in the centre (Lazio (5), Marche (7), Tuscany (6), Umbria (5)), and in the south (Abruzzo (5), Basilicata (5), Campania (5), Apulia (5), Sardinia (5), Sicily (5)) of the country.

In addition, a sample of authentic grape must (*δ*^13^C_GLUCOSE_ −29.1‰ and *δ*^13^C_FRUCTOSE_ −27.6‰) was added with growing percentages (from 0 to 40%) of two different sugar adulterants: glucose (−10.9‰) and fructose (−10.7‰) from C4 plants.

A dilution of 1:500 in milli-Q water was performed for must samples. The standard solution used to calculate the instrument drift and to correct the values of the samples had a concentration of 0.4 mg/mL in both D- (+)-Glucose and D- (−)-Fructose. All samples were filtered through a 0.22 µm filter (Millipore Millex-GV, PVDF 0.22 µm) prior to LC analysis.

### 3.3. LC-IRMS

To carry out the analyses, an LC (UHPLC UltiMate 3000, Dionex) linked with an IRMS mass spectrometer (Delta V Plus, Thermo Scientific, Bremen, Germany) via the LC-Isolink interface (Thermo Scientific, Bremen, Germany) was used. The UHPLC was equipped with a single pump, a 120 positions autosampler and an RS column compartment that made it possible to keep the column at a constant temperature of 80 °C during the analysis. A Rezex RCM-Monosaccharide Ca + 2 column 300 × 0.8 mm (Phenomenex, Torrance, CA, USA) was used. The injected volumes were 4 µL for the standard solution and 10 µL for the samples. The fluxes were set at 400 µL/min for the LC system and at 20 µL/min for each of the two reagents (acid and oxidant). In the LC IsoLink, the liquid phase is not removed from the sample prior to oxidation but the sample is oxidized while still within the aqueous solution, and only afterwards the CO_2_ is separated from the liquid phase for isotopic analysis. This process is quantitative and fractionation-free in the LC IsoLink. The eluted solution is mixed with ammonium persulfate (15% in mass), oxidative solution, and orthophosphoric acid solution (2.5% in volume) via a three-way Valco valve. Both are pumped separately and added to the mobile LC phase. All individual organic compounds eluting from the LC column are oxidized quantitatively into CO_2_ within this mixture when passing through a heated reactor. In a downstream degassing unit, the CO_2_ is removed from the liquid phase and entrained into a stream of helium. The individual CO_2_ peaks in helium are subsequently dried in an on-line gas drying unit (Nafion^®^) with an external helium flux of 50 mL/min and then admitted to the IRMS via an open split interface. Since the liquid phase is not vaporized at all, inorganic buffers, e.g., phosphate buffers, have no influence on the operation of the interface, and cannot accumulate anywhere in the system. Eluent and reagent solutions are previously de-aerated by bubbling helium for half an hour and kept under helium flux during experiments to avoid any atmospheric CO_2_ contamination. In order to prevent any ammonium persulfate crystallization in the tubing, the system is rinsed with water as soon as unoccupied and left under water flux overnight [10].

### 3.4. Stable Isotope Ratio Analysis and Data Correction

According to the IUPAC protocol [31], the ^13^C/^12^C values are expressed in the delta scale (*δ*‰), against the international standards V-PDB (Vienna-Pee Dee Belemnite) according to the Equation (1):(1)$$\delta_{ref}{(}^{i}E{/}^{j}E,sample)=\left[\frac{R{(}^{i}E{/}^{j}E,\;\;\;sample)}{R{(}^{i}E{/}^{j}E,\;\;\;ref)}\right]-1$$
where *ref* is the international measurement standard, *sample* is the analysed sample, and *^i^E/^j^E* is the isotope ratio between heavier and lighter isotopes. The delta values were multiplied by 1000 and expressed in units “per mil” (‰). Each sample was analysed in triplicate.

The *δ*^13^C values were corrected for the instrumental drift and calculated against working standard materials (D-(+)-Glucose (−20.0 ± 0.2‰) and D-(−)-Fructose) (−26.5 ± 0.2‰) and injected in triplicate at the beginning and at the end of each analytical sequence. The working standards were calibrated using an EA-IRMS (Flash 1112, Thermo Scientific, Bremen) against the international standards NBS-22 fuel oil (IAEA-International Atomic Energy Agency, Vienna, Austria), IAEA-CH-6 sucrose, and USGS 40 (U.S. Geological Survey, Reston, VA, USA). We did not use a calibration curve for *δ*^13^C as suggested by IUPAC protocol [31] because, as we used a single standard with a value similar to that of the samples, the data determined using a single anchoring point or two–three anchoring points were not significantly different.

### 3.5. Statistical Analysis

To perform the statistics, Statistica 13.3.704.1 was used. Once data normality was assessed, a one-way ANOVA was performed to check statistical differences between groups (Tukey post hoc test, *p* < 0.01).

## 4. Conclusions

The use of the LC-IRMS technique has proven to be a powerful alternative for the analysis of the isotopic composition of grape must sugars, providing repeatable results. A good discrimination between grape samples from different Italian regions was carried out according to the *δ*^13^C of glucose and fructose. Statistically higher values have been found in central and southern Italy regions than in northern ones (*p* < 0.01). Once widened, this preliminary dataset might represent the basis of a databank, which will make it possible to discriminate between Italian regions.

By using the LC-IRMS technique, the addition of exogenous sugars, such as fructose and glucose from C4 photosynthetic cycle plants, is easily detectable as it modifies the *δ*^13^C of the individual sugars. This method might be proposed as a quicker and cheaper alternative for measuring the *δ*^13^C of the samples used to build the official must and wine database prescribed for all member countries of the European Community. It would, therefore, be desirable to amend the present legislation, which currently recognizes the OIV MA AS-312-06 method as the only one for analysing the *δ*^13^C of ethanol obtained after the fermentation and distillation of musts.

## Figures and Tables

**Figure 1 molecules-28-01411-f001:**
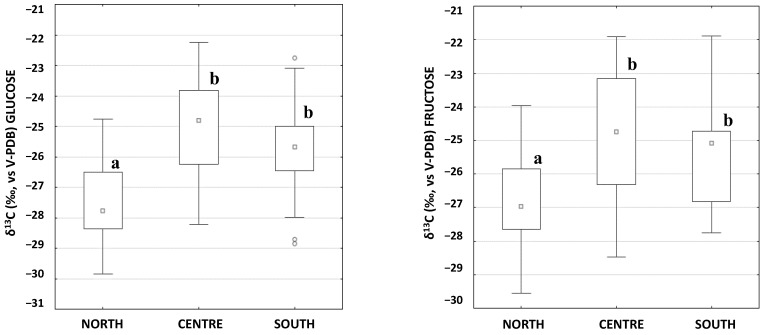
Box-Whisker plot of the glucose and fructose LC-IRMS analysis divided by the origin of the sample (north, centre or south Italy). Different letters identify statistically different groups according to the ANOVA (*p* < 0.01).

**Figure 2 molecules-28-01411-f002:**
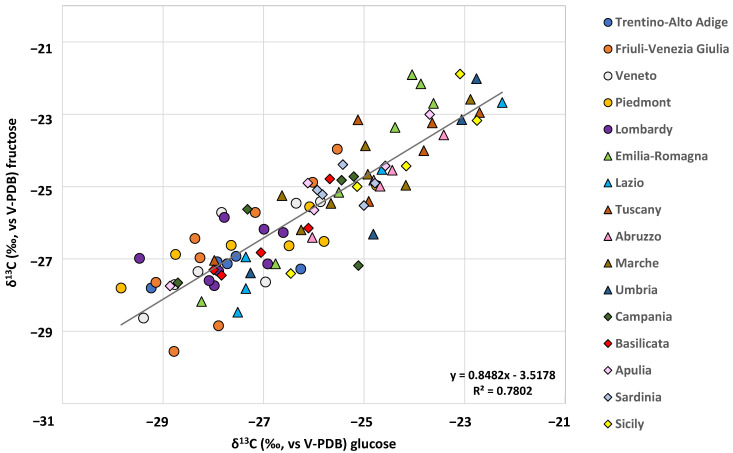
Correlation between *δ*^13^C_FRUCTOSE_ and *δ*^13^C_GLUCOSE_.

**Figure 3 molecules-28-01411-f003:**
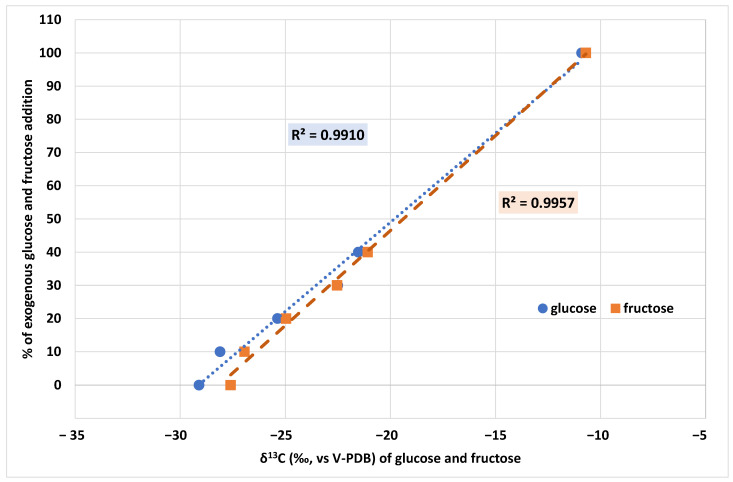
Variation of δ^13^C_GLUCOSE_ and *δ*^13^C_FRUCTOSE_ of an authentic grape must sample by the addition of growing % of exogenous glucose and fructose.

**Table 1 molecules-28-01411-t001:** Within-day repeatability and between-day repeatability of the method. The mean *δ*^13^C for glucose and fructose are reported together with the RSD (%). The ^13^C/^12^C values are expressed in the delta scale (*δ*‰), against the international standards V-PDB (Vienna-Pee Dee Belemnite).

	*δ*^13^C (‰, vs. V-PDB) Glucose	*δ*^13^C (‰, vs. V-PDB) Fructose		*δ*^13^C (‰, vs. V-PDB) Glucose	*δ*^13^C (‰, vs. V-PDB) Fructose
1	−27.5	−27.7	Day 1	−27.3	−27.4
2	−26.6	−26.5	Day 2	−26.1	−27.8
3	−27.3	−26.9	Day 3	−27.1	−27.1
4	−27.6	−27.0			
5	−28.1	−28.6			
6	−28.2	−29.0			
7	−28.9	−26.9			
8	−27.9	−27.4			
Mean (‰)	−27.8	−27.5	Mean (‰)	−26.8	−27.5
Whitin-day RSD (%)	2.5	3.2	Between-day RSD (%)	2.5	1.3

## Data Availability

The data presented in this study are available in the present article.

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
