# Peer review of "Authentication and Geographical Characterisation of Italian Grape Musts through Glucose and Fructose Carbon Isotopic Ratios Determined by LC-IRMS"

_molecules, 2023, doi:10.3390/molecules28031411_

Round 1
Reviewer 1 Report
The Authors must revise and correct the references 22 and 23 as 26 and 27, because they are repeated in References. The final reference numbers must be corrected in the manuscript.
In the Table 1, the Authors must incorporate in the legend " 13C/12C values are expressed in the delta scale (δ‰), against the international standards V-PDB (Vienna-Pee Dee Belemnite) for carbon".
Author Response
The Authors must revise and correct the references 22 and 23 as 26 and 27, because they are repeated in References. The final reference numbers must be corrected in the manuscript.
We corrected the references as requested.
In the Table 1, the Authors must incorporate in the legend " 13C/12C values are expressed in the delta scale (δ‰), against the international standards V-PDB (Vienna-Pee Dee Belemnite) for carbon".
We improved Table 1 as requested.
Reviewer 2 Report
This manuscript addresses authentication of Italian grape by using liquid chromatography coupled with isotope ration mass spectrometry. The authors analyze isotope ratio of glucose and fructose in Italian grapes and estimate robustness by within-day repeatability and between-day reproducibility. This manuscript provides the wide usages of this technique, however, minor revisions will be necessary for publication
L97. Table 1. Please show “Sr” and “SR in table title. I suppose that it means standard deviation, if so delta13 C of glucose in between-day might be 0.6. Please check it.
L101. Please move to bottom the figure.
L105. The authors mentioned difference of isotope ration. Please show briefly the reasons.

Author Response
This manuscript addresses authentication of Italian grape by using liquid chromatography coupled with isotope ration mass spectrometry. The authors analyze isotope ratio of glucose and fructose in Italian grapes and estimate robustness by within-day repeatability and between-day reproducibility. This manuscript provides the wide usages of this technique, however, minor revisions will be necessary for publication
L97. Table 1. Please show “Sr” and “SR” in table title. I suppose that it means standard deviation, if so delta13 C of glucose in between-day might be 0.6. Please check it.
Following the suggestion of Reviewer 3, we changed the standard deviation to the relative standard deviation (RSD).
L101. Please move to bottom the figure.
We moved the description of Figure 1 from the top to the bottom of the Figure itself.
L105. The authors mentioned difference of isotope ration. Please show briefly the reasons.
We inserted a concise explanation of the different isotopic ratios found in different areas of Italy. Some references have been also included.
Reviewer 3 Report
Manuscript Number: Molecules-2144398
Title: Authentication and geographical characterisation of Italian grape musts through glucose and fructose carbon isotopic ratios determined by LC-IRMS
Comments to the authors
The authors propose an alternative method for the authentication and geographical characterisation of Italian grape musts based on δ13C isotopic analysis of their major sugars by liquid chromatography coupled with isotope ratio mass spectrometry (LC-IRMS). The instrumentation used is suitable for this purpose and the topic addressed seems to be of interest to the journal readers. Nevertheless, there are several points that are not clear in this version of the manuscript and thus need to be better clarified. I understand that this manuscript can be considered for publication after making major revisions.
The following points should be covered in detail in the revised version:
1) The Graphical Abstract was not included in this manuscript.
2) In the Abstract section, the findings of this research including relevant experimental data should be described in detail, as well as the advantages of this method over other methods in the literature must be highlighted.
3) In the Introduction section, the difficulties of the current methods for the authentication of grape musts addressed in lines 54-59 must be supported by their corresponding references. Please include references at these sentences.
4) The sentences of lines 78-85 must be included in a single paragraph.
5) The Results and discussion section should be placed after Materials and methods section.
6) The standard deviation must be expressed as a relative standard deviation in percentage (%).
7) The terms Sr and SR are confusing in Table 1, please clarify.
8) In the current working conditions, the term reproducibility must be changed to intermediate precision.
9) The validation data must be compared with the reference values established by the corresponding validation guide to verify that the analytical performance criteria are fulfilled.
10) The figure legend should be placed below the figures and evenly throughout the manuscript.
11) Please use 4 decimal places for R2.
12) Figure 3 should be included after being cited in the text.
13) In the Samples section, it should be including how many grapes must samples correspond to each of the 16 sampling regions.
14) Substitute Ca+2 for Ca2+.
15) I am not sure if by not using a calibration curve for δ13C as suggested by the IUPAC protocol, the results described in this work can be trusted. To overcome this issue, it is recommended to compare the results reported here with those obtained by a standard method and estimate significant differences using the Student's t-test.
16) The explanation of the statistical data in relation to the experimental data throughout the manuscript should be improved.
17) The advantages of this method with respect to other methods in the literature should be discussed in the Results and discussion section.
18) It is recommended to include more references, as well as more recent references.
19) The style of the references must be checked in detail to avoid errors introduced automatically by the software.
20) Based on findings of this research, authors are encouraged to use a classification approach that can help isotopic carbon (δ13C) measurements through a prediction model aimed at easier discriminate the provenance of the must samples. If this is not possible due to the data over-fitting that would be obtained by using a reduced number of samples per class (provenance), it is recommended that this work be published as a communication.
Author Response
The authors propose an alternative method for the authentication and geographical characterisation of Italian grape musts based on δ13C isotopic analysis of their major sugars by liquid chromatography coupled with isotope ratio mass spectrometry (LC-IRMS). The instrumentation used is suitable for this purpose and the topic addressed seems to be of interest to the journal readers. Nevertheless, there are several points that are not clear in this version of the manuscript and thus need to be better clarified. I understand that this manuscript can be considered for publication after making major revisions.
The following points should be covered in detail in the revised version:
1) The Graphical Abstract was not included in this manuscript.
We are sorry if the abstract might be not visible in the main manuscript. Nevertheless, it has been uploaded the during the submission procedure.
2) In the Abstract section, the findings of this research including relevant experimental data should be described in detail, as well as the advantages of this method over other methods in the literature must be highlighted.
We improved the Abstract as requested.
3) In the Introduction section, the difficulties of the current methods for the authentication of grape musts addressed in lines 54-59 must be supported by their corresponding references. Please include references at these sentences.
We included supporting references and improved the paragraph as requested.
4) The sentences of lines 78-85 must be included in a single paragraph.
Hoping to have well interpreted your suggestion, we decided to gather lines 76-77 and 78-85 in a single paragraph, which should figure as a conclusive part of the Introduction.
5) The Results and discussion section should be placed after Materials and methods section.
In writing the manuscript, we followed the order suggested in Molecules Guidelines for authors.
6) The standard deviation must be expressed as a relative standard deviation in percentage (%).
We changed the standard deviation to the relative standard deviation both in Table 1 and through the test, as requested.
7) The terms Sr and SR are confusing in Table 1, please clarify.
We changed the standard deviation SR to the relative standard deviation RSD, as requested at point 7.
8) In the current working conditions, the term reproducibility must be changed to intermediate precision.
We agree and we changed the term reproducibility in the test.
9) The validation data must be compared with the reference values established by the corresponding validation guide to verify that the analytical performance criteria are fulfilled.
Thanks for your suggestion. Trying to provide a comprehensive answer to your comment, we have decided to refer to the specifications of the LC-IRMS manufacturer. In particular, the validation data we discussed in “Method Validation” section are consistent with those obtained by the manufacturer in a study of honey sugars.
More information is described in the Application note LC-IRMS: Authenticity Control of Honey Using the Thermo Scientific LC IsoLink LC-IRMS, available at (https://www.thermofisher.com/order/catalog/product/IQLAAEGAATFAETMAHX?gclid=Cj0KCQiAtvSdBhD0ARIsAPf8oNkNHvRNfeUV9MHCtW3cWW1v0HX826AE4Vryx_kagIHRgh1E0rBgzgaAjBpEALw_wcB&cid=E.23CMD.DL103.12913.01&ef_id=Cj0KCQiAtvSdBhD0ARIsAPf8oNkNHvRNfeUV9MHCtW3cWW1v0HX826AE4Vryx_kagIHRgh1E0rBgzgaAjBpEALw_wcB:G:s&s_kwcid=AL!3652!3!334040549241!p!!g!!lc-irms).
Furthermore, they are not dissimilar to what reported by El Hawari K et al. "Evaluation of honey authenticity in Lebanon by analysis of carbon stable isotope ratio using elemental analyzer and liquid chromatography coupled to isotope ratio mass spectrometry". J Mass Spectrum. 2021 Apr 15;56(6):e4730. doi:10.1002/jms.4730
10) The figure legend should be placed below the figures and evenly throughout the manuscript.
We moved the description of Figure 1 from the top to the bottom of the Figure itself.
11) Please use 4 decimal places for R2.
We modified the R2 in Figure 2 and 3 as requested.
12) Figure 3 should be included after being cited in the text.
We moved Figure 3, as requested.
13) In the Samples section, it should be including how many grapes must samples correspond to each of the 16 sampling regions.
We inserted the number of samples considered for each region, in Samples section.
14) Substitute Ca+2 for Ca2+.
We would prefer to keep the name of the LC column as it is indicated in the Phenomenex catalogue.
15) I am not sure if by not using a calibration curve for δ13C as suggested by the IUPAC protocol, the results described in this work can be trusted. To overcome this issue, it is recommended to compare the results reported here with those obtained by a standard method and estimate significant differences using the Student's t-test.
As suggested, two pure and commercial glucose samples with different isotopic content were analyzed both directly by means of the EA-IRMS technique (correcting with respect to two international reference standards IAEA CH6 and USGS40) and by dilution and LC-IRMS analysis, correcting with respect to the D-( +)-Glucose used in the present work (-20.0±0.2‰). The values found (-10.9‰ vs -10.8‰ and -23.7‰ vs -23.9‰) are not statistically dissimilar considering the estimated uncertainty.
Other tests are reported in the first part of chapter 2.1.
16) The explanation of the statistical data in relation to the experimental data throughout the manuscript should be improved.
We improved the description of the statistical data in relation to the experimental data in the Results and Discussion part.
17) The advantages of this method with respect to other methods in the literature should be discussed in the Results and discussion section.
We would prefer to keep the discussion of the advantages of this technique in the Introduction part, that we highly improved by adding more references and more information about the comparison between LC-IRMS and other analytical approaches.
18) It is recommended to include more references, as well as more recent references.
We substituted some references with more recent ones and included more literature.
19) The style of the references must be checked in detail to avoid errors introduced automatically by the software.
We carefully checked the references looking for errors, as requested.
20) Based on findings of this research, authors are encouraged to use a classification approach that can help isotopic carbon (δ13C) measurements through a prediction model aimed at easier discriminate the provenance of the must samples. If this is not possible due to the data over-fitting that would be obtained by using a reduced number of samples per class (provenance), it is recommended that this work be published as a communication.
Thank you for the constructive suggestion. Unfortunately, even though the number of total samples is more than enough to make a comparison among different Italian areas, we have still few measurements for each region to be able to satisfy your request
Round 2
Reviewer 3 Report
Manuscript ID: molecules-2144398
Authentication and geographical characterisation of Italian grape musts through glucose and fructose carbon isotopic ratios determined by LC-IRMS
The authors have provided a point-by-point response to the reviewer’s comments. Nevertheless, some important points to improve the scientific quality of this research were not considered due to lack of data. For example, the authors state that the total number of samples is more than enough to make a comparison between different Italian areas, but that they have still few measurements for each region, so implementing a discrimination model is not possible here. This approach was to be fundamental since the present work focuses on the authentication of grape musts.
Note: Some text editing issues that must be corrected if this manuscript is accepted for its publication are detailed below:
It is clearly noticeable that there are missing spaces between some words and there are repeated punctuation marks. It is also noticeable the inclusion of new phrases in substitution of initial phrases that had to be eliminated, which brings confusion to the reader. I think there was a lack of care in this regard.
Finally, considering the modifications to the text based on the reviewers' suggestions, I think the English edition should be checked.
Author Response
Reviewer 3
Manuscript ID: molecules-2144398
Authentication and geographical characterisation of Italian grape musts through glucose and fructose carbon isotopic ratios determined by LC-IRMS
The authors have provided a point-by-point response to the reviewer’s comments. Nevertheless, some important points to improve the scientific quality of this research were not considered due to lack of data. For example, the authors state that the total number of samples is more than enough to make a comparison between different Italian areas, but that they have still few measurements for each region, so implementing a discrimination model is not possible here. This approach was to be fundamental since the present work focuses on the authentication of grape musts.
We inserted comments throughout the test to clarify that this is an explorative study carried out by considering a preliminary dataset that will have to be improved to become a databank. We also highlighted that the aim of this work was to prove the applicability of the proposed method and to first apply it on Italian musts.
Note: Some text editing issues that must be corrected if this manuscript is accepted for its publication are detailed below:
It is clearly noticeable that there are missing spaces between some words and there are repeated punctuation marks. It is also noticeable the inclusion of new phrases in substitution of initial phrases that had to be eliminated, which brings confusion to the reader. I think there was a lack of care in this regard.
A clean version of the manuscript, corrected according to the last suggestions and to the English revision, has been uploaded on Molecules upload system.
Finally, considering the modifications to the text based on the reviewers' suggestions, I think the English edition should be checked.
A native English speaker employed by the company EUROSTREET SOCIETÀ COOPERATIVA (info@eurostreet.it) revised the text.